ecology

dispersal, vector, *Pontodrilus litoralis*, earthworm, intertidal

**Author for correspondence:**
Keryea Soong
e-mail: keryea@g-mail.nsysu.edu.tw

# How to cross the sea: testing the dispersal mechanisms of the cosmopolitan earthworm *Pontodrilus litoralis*

## Shiao-Yin Chen, Chia-Hsuan Hsu and Keryea Soong

Department of Oceanography, National Sun Yat-sen University, Kaohsiung, Taiwan

C-HH, 0000-0002-5128-5902; KS, 0000-0002-4371-7103

Dispersal capability often decides the geographical distribution and long-term success of a species. In this investigation, *Pontodrilus litoralis*, a widely distributed species along shores throughout mid- and low latitudes of the world, was investigated. We tested three hypotheses explaining its dispersal: helped by humans, transported by birds and carried by currents. Although the earthworms seemed to be associated with artificially planted wind-breaking woods and mangroves along the west coast of Taiwan, they were also found on isolated beaches in the Pescadores Islands without such plantings. They are approximately 2 mm wide, making them too small for use as fishing bait. These two mechanisms invoking human help were not supported. In a laboratory experiment, we moved the earthworms to the plumage of various body parts of pigeons, and they dropped off or died within a short time, a result incompatible with the bird hypothesis. The earthworms stayed alive and grew when immersed in freshwater or seawater for at least a month. They also survived on floating wood in an *in situ* experiment lasting approximately two months. Thus, the current hypothesis was the only one we were unable to falsify; driftwood and perhaps wooden vessels could provide both food and transport on long journeys. Wood boats exist for a short time on an evolutionary time scale, but it may be long enough to disperse the earthworm around the world. The phase-out of wood boats, thus, may start the divergence of *P. litoralis* populations around the world.

## 1. Introduction

Dispersal away from natal habitats is crucial for all species because suitable habitats do not last forever. Kin selection, inbreeding avoidance and uneven spatial–temporal patterns of resource distribution are among the likely selective mechanisms

at the individual level [1]. At least three types of mechanisms have been invoked to explain the disjunct spatial distribution of a species: (i) self-mobility [2]; (ii) passive carrying by wind, water or other vectors [3] and (iii) human activities [4,5]. In addition, plate tectonics is a possible mechanism causing patterns in biogeography, but it occurs on a time scale much longer than the generation time of organisms. Vicariance is thus often considered an alternative hypothesis to dispersal, rather than part of it, when explaining the distribution of species [6].

The field of island biogeography started out to predict the species richness of islands on which virtually all species originated elsewhere [7]. Many tests have been devised to test the possible mechanisms of dispersal [8,9]. How a species departs its natal habitats and settles in a new isolated ecosystem is easy to infer in some species but remains challenging to infer in others [10].

*Pontodrilus litoralis*, a euryhaline earthworm, is one such example. It was first reported to be present in tropical coasts in South India, Australia and New Zealand [11] and was later found along the coasts of the Mediterranean, North America, the Caribbean, Taiwan, Pacific islands, Thailand, Malaysia and the Sea of Oman [12–25]. The wide and disjunct distribution, especially on oceanic islands, suggests the existence of an effective long-distance dispersal mechanism. Although the populations in different regions of the world were initially assigned different names [26,27], no morphological difference exists among these populations, and they are considered the same species [4,16,21,28]. The following questions require answers: How did this earthworm disperse to numerous islands in three oceans? Why have they not diverged?

The habitats of *P. litoralis* are beaches, brackish mud flats, intertidal mangrove swamps and even wooden piers [4,29]. Considerable variation exists in the salinity of their habitats [30].

Human activities such as gardening and fishing are known to disperse many terrestrial earthworms, including *Pontoscolex corethrurus* and *Eudrilus eugeniae* [29,31–33]. For *P. litoralis*, the planting of wind-breaking woods along coasts is a dispersal opportunity. In fact, planting or transplanting is considered one of the common mechanisms through which seashore organisms are dispersed regardless of whether they themselves are the targets [9,33]. *Pontodrilus litoralis* individuals are easily carried in soil and sand when plants are moved and replanted.

In addition, birds may disperse seeds and even animals, either by carrying them in their digestive tracts or on their body surface [1,34–36]. Earthworms are prey of numerous shore birds. The worms are unlikely to survive in the digestive systems of birds without protection of any kind [4]. Nevertheless, birds are likely to spend much time near earthworm habitats; furthermore, they travel a long distance within a short time, and thus, their potential to serve as vectors deserves investigation. Methods through which earthworms can travel on birds may include stowing away beneath feathers [4] or feet [37], either as worms or fertilized eggs. However, prior to that stage, the earthworm must board the vector bird somehow. The likelihood of this could be tested in a laboratory.

Ocean currents are a common dispersal agent and represent a reasonable hypothesis for this particular earthworm [4,38]. For example, the earthworm's requirement of moisture maintenance, a challenge under both the human and bird hypotheses, is easily solved in seawater. Because of the long duration required to travel by currents, both food and shelter must be provided by a potential vector. Herein, we consider three possible candidates: driftwood, seagrass macrodetritus and wood boats. All three are present on the beaches of islands and are known to be associated with *P. litoralis* in nature (e.g. on Dongsha Island of Taiwan, according to our personal observations). All three may be pushed out to sea, which would indicate departure (the first step in dispersal)—seagrass macrodetritus when the wind direction is favourable, beached driftwood during typhoons and wood boats when moved to the water. An early study revealed that earthworms have the potential to travel by ocean currents; they survived 4 days under salinity of 2.5–30% [30]. However, one study observed that *P. litoralis* may be irritated by seawater [16]. Because dispersal through drifting in currents requires a long duration of transfer (the second stage of dispersal), whether this earthworm can survive long periods of immersion in seawater must be tested.

In Taiwan, *P. litoralis* is considered an alien species (from the Taiwan Biodiversity Information Facilities: http://taibif.tw/zh/namecode/402726); however, no relevant evidence exists to judge whether they arrived through human activities or natural mechanisms. Thus, we designed our study to test whether the species has the ability to disperse over long distances without human intervention.

The genus *Pontodrilus* actually has a few other known species, including *P. longissimus* found in the coasts of Thailand and Malaysia; it is often sympatric with *P. litoralis* [25]. Nevertheless, only *P. litoralis* achieved wide distribution in the world. It is certainly possible that some local populations may have diverged and even speciated, but the dispersal mechanism remains a valid question [39].

In this investigation, observations and experiments were designed to test various hypotheses explaining how the intertidal earthworm *P. litoralis* could be so widely distributed globally. The three stages of dispersal—namely departure (or emigration), transfer (or transience) and settlement (or

**Table 1.** Hypotheses of *P. litoralis* dispersal.

| dispersal stage | hypothesis | | |
| --- | --- | --- | --- |
| | human | bird | current |
| departure (emigration) | collected as bait or along with sand | stowed away or attached to birds | Wood and boats on riverbanks and on beaches or docks. Seagrass macrodetritus blown out to sea. |
| transfer (transience) | under human control, such as in pots, plant nurseries and ballasts | long-distance dispersal could be accomplished in a short time | Hidden in cracks and crevices of wood and boats. On floating seagrass macrodetritus. |
| settlement (immigration) | during planting and unloading of ballast sand; through fishermen | migrating birds arriving on new islands or in new habitats | Beached/docked wooden trunks and boats. Seagrass macrodetritus landing. |

immigration) [1,40–42]—could be independently evaluated in our study (table 1). We assessed the first two stages: that is, the ability to cling to vectors in the departure stage, and survival and growth during the transfer stage. The human hypothesis was tested by determining whether the occurrence of *P. litoralis* was positively associated with human plantation of wind-breaking woods along the coast. The bird hypothesis was investigated by studying how the earthworm could access birds and, once 'boarded', whether the earthworms could remain stowed under their plumage. The current hypothesis was tested by investigating whether the earthworms can survive and grow on either wood or seagrass macrodetritus in seawater. Finally, we compared hypotheses involving wood (i.e. driftwood and wood boats) in the sea.

## 2. Material and methods

### 2.1. Distribution in Taiwan and surrounding islands

The shores along the east coast of Taiwan (22–25°N, 121.5°E) are mostly rocky, with a few beaches scattered near estuaries. The west coast of Taiwan (22–25°N, 120–121.5°E) is mostly sedimentary with artificially planted wind-breaking woods, and some mangroves and estuaries are scattered between beaches. Using Google Maps, we located potential beaches in Taiwan and its surrounding islands to survey for *P. litoralis* between September 2011 and September 2013 (figure 1). Many beaches have artificially planted woods consisting of swamp oaks, also known as horsetail trees (*Casuarina equisetifolia*), which have the ability to resist wind, salty water and drought; *P. litoralis* could be found in damp parts of these woods, often close to the shore (K.S. 2011, personal observations). It is possible that these euryhaline earthworms were transported there when the trees were first planted.

The Pescadores Islands (23.4°N, 119.5°E) are an archipelago composed mostly of basalt, although scattered beaches exist between rocky shores. Rocky shores are not suitable habitats for *P. litoralis*; thus, the isolated beaches are effectively ecological islands that intertidal earthworms can independently colonize, presumably through one of the hypothesized dispersal mechanisms. We surveyed beaches on the Pescadores Islands using a standard method. At each beach, we selected the most appropriate spot on the basis of the water level (near the high water mark and immersed at high spring tide), the accumulation of organic materials (the more the better) and oxygenated sediment, and then checked a 50 cm square down to 15 cm below the surface. The presence or absence of the earthworms was noted for each beach. The sampling was repeated at approximately 200 m intervals if the beach was longer than that. The beaches were categorized as having or not having swamp oak woods within 50 m.

Orchid Island (22.0°N, 121.5°E) and Green Island (22.6°N, 121.5°E) to the southeast of Taiwan are two oceanic islands with scattered beaches along the coasts of fringing reefs. They were surveyed in a similar manner, although much fewer sites were considered suitable for the earthworms. Dongsha Island (20.7°N, 116.7°E) to the southwest of Taiwan is an atoll composed totally of coral sand.

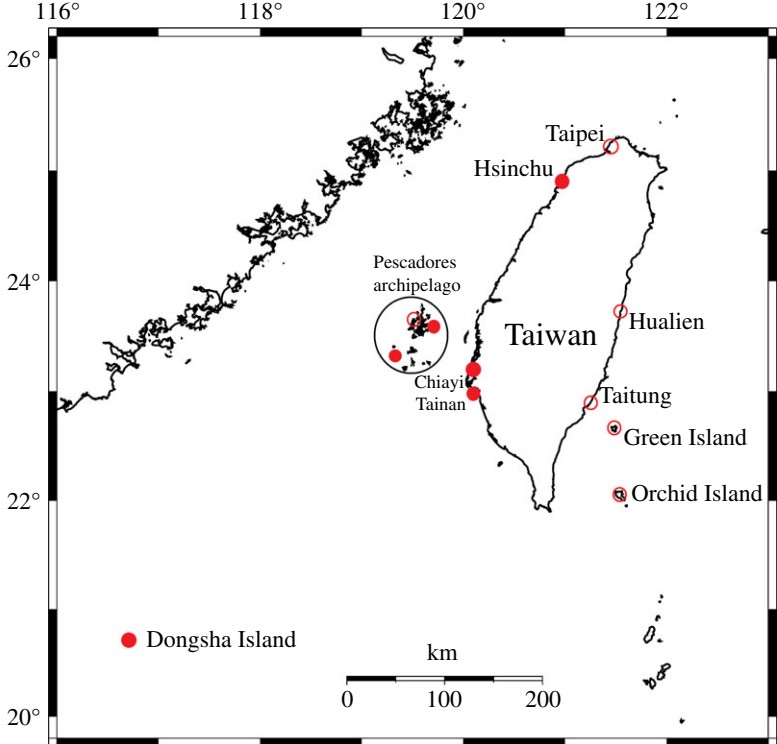

**Figure 1.** Survey of *P. litoralis* in Taiwan. Solid: presence, blank: absence.

## 2.2. Vertical distribution

Using $15 \times 15$ cm quadrats, we counted the number of earthworms in every 3 cm thick layer of seagrass or sand down to approximately 25 cm below the interface of seagrass macrodetritus and sand in the upper intertidal zone of Dongsha Island during February–March 2012 and January–February 2013. At other sites, only the presence or absence of data was retained because of the low numbers of earthworms and limited distribution near the surface.

## 2.3. Clinging experiments (departure and transfer stages)

All the earthworms used for experimentation were collected on the north coast of Dongsha Island; a collection permit was obtained from the Marine National Park. The earthworms actively avoid light, and this property was used to concentrate them using transparent plastic bags or conversely to rid seagrass macrodetritus of the earthworms.

To travel using currents or birds, *P. litoralis* must stay on or attach to vectors for long periods because the earthworm itself sinks in seawater. This capability was tested in three candidate vectors.

### 2.3.1. Seagrass leaves

Seagrass macrodetritus accumulates in large quantities (e.g. 50 cm in thickness at Dongsha Island) on the shore, seasonally. When the wind direction changes, such as to the southwest, seagrass macrodetritus on the north coast is blown out to sea. This provides numerous opportunities for sea-going objects. Earthworms, if able to cling to seagrass, might use it to travel the ocean. To test the feasibility of this, a total of 30 earthworms were placed in three bags, with each bag containing 1 kg of seagrass macrodetritus (mostly *Cymodocea rotundata* and *Thalassia hemprichii*), and kept in darkness for half a day. Then, they were placed in a 2 l beaker filled with seawater to test whether they stayed on the floating seagrass or were unable to secure to it and sank to the bottom.

### 2.3.2. Wood

Beached wood is common in many of the locations, and *P. litoralis* can be found in its cracks and crevices. We tested whether the earthworms could stay with the wood when it was floating in water. Wooden

blocks of approximately $3 \times 3 \times 1.5$ cm were made on the campus of National Sun Yat-sen University from driftwood collected from beaches. Then, each of the 30 earthworms was kept with a block for half a day to settle. During the test, these 30 blocks were individually placed in 2 l beakers filled with seawater. After 30 min, the earthworms that had not secured to the wood and had sunk to the bottom of the beakers were counted.

### 2.3.3. Birds

Shore birds wade on beaches, and their proximity to earthworm habitats lends some credence to the hypothesis that they could be a vector. An earthworm was planted on one of five positions—namely the prothorax, mantle, tertials, tail and abdomen—on two pigeons. The earthworms were placed inside the plumage near the skin. The presence and condition of each earthworm was checked at 5, 10, 60 and 120 min. The experiment on each pigeon was performed 15 times. The use of the pigeons was approved by the Institutional Animal Care and Use Committee (IACUC) of National Sun Yat-sen University.

## 2.4. Seawater versus freshwater experiment (transfer stage)

To test whether *P. litoralis* could live in a freshwater or seawater environment for a long time, two experiments were conducted. In the first experiment, we immersed seagrass macrodetritus (mostly *T. hemprichii*) in either seawater or freshwater to keep it moist and then cultured the earthworms in these two treatments. A total of 45 earthworms (30 in seawater; 15 in freshwater) were individually weighed after being blotted dry. Then, each was placed in a cloth bag made of synthetic fibre, along with 15 g (air dry weight) of seagrass macrodetritus. Each bag was immersed either in seawater (salinity 30 psu) or freshwater, depending on the group each belonged to, for 10 min week$^{-1}$ to maintain moisture, and all the bags were kept in dark containers the rest of the time. The survival of each was examined at two-week, one-month and two-month intervals after the start of the experiment. Individual earthworms were weighed again one month later and at the end of the experiment.

## 2.5. Field survival (transfer stage)

To test whether *P. litoralis* could live with wood or seagrass macrodetritus for a long period in seawater, we used black synthetic cloth to make bags with three treatments: wooden block measuring $3 \times 3 \times 1.5$ cm, 20 g of the seagrass debris, or a piece of black synthetic cloth measuring $15 \times 4$ cm placed in the bags. The piece of wood was collected from a beach on Dongsha Island and cut to the appropriate size. The wooden blocks and seagrass macrodetritus were preconditioned in seawater for two weeks before the experiment. Each treatment was performed 30 times; thus, a total of 90 bags were used. All the bags with earthworms were deployed in a small seawater lagoon with an opening to the sea on Dongsha Island.

# 3. Results

## 3.1. Spatial distribution

*Pontodrilus litoralis* occurred on the west coast of Taiwan, which is dominated by sedimentary beaches, but not on the beaches of the east coast, which are mostly rocky shores and rubble.

In the Pescadores Islands, where suitable habitats of *P. litoralis* are scattered, we completed more independent observations. Four sites in this survey, three mentioned in the literature [16] and Wan'an (23.36°N, 119.49°E), were found to have the earthworm; however, none of these sites have wind-breaking woods nearby. In addition, no earthworms were discovered on 60 other isolated beaches that were checked, and only one of these sites had wind-breaking woods in the vicinity. The potential association between earthworm presence and wind-breaking woods was not substantiated in the Pescadores (table 2).

*Pontodrilus litoralis* occurred on the beaches of Dongsha Island and was especially abundant on the north coast where seagrass macrodetritus accumulated near the high water mark. They were not found near low or intermediate intertidal levels where the water content of the sediment was high. The species was not found on the beaches of Green Island or Orchid Island.

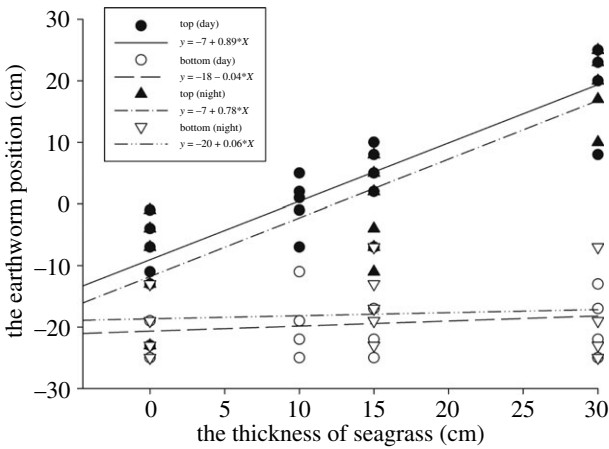

**Figure 2.** Vertical ranges of the *P. litoralis* earthworms on Dongsha Beach with seagrass piles of various thickness during the day and night. No difference between day and night was discovered. The thicker the seagrass pile ($X$), the thicker the range of the vertical distribution of the earthworm; $Y = 11 + 0.89X$, $X$: the thickness of the seagrass pile; $W$: range of the vertical distribution, $W = Y1 - Y2$ or $Y3 - Y4$; $N = 84$ columns of sampling. '0' in $Y$ indicates the interface between sand and a seagrass pile. Upper limit (day): $r = 0.90$, $p < 0.001$; $Y1 = -7 + 0.89 \times X$. Lower limit (day): $r = -0.08$, $p = 0.62$; $Y2 = -18 - 0.04 \times X$. Upper limit (night): $r = 0.85$, $p < 0.001$; $Y3 = -7 + 0.78 \times X$. Lower limit (night): $r = 0.15$, $p = 0.33$; $Y4 = -20 + 0.06 \times X$.

**Table 2.** Summary of sites with presence or absence of *P. litoralis* earthworms and wind-breaking woods in the Pescadores Islands ($p > 0.05$, Fisher's exact test).

| | windbreaker woods presence | |
| --- | --- | --- |
| earthworm presence | yes | no |
| yes | 0 | 4 |
| no | 1 | 60 |

## 3.2. Vertical distribution

On Dongsha Island, the earthworms started to be found 3 cm below the surface regardless of whether the substrate was seagrass macrodetritus or sand, and they could be found all the way down to a maximum of approximately 25 cm below the sand surface. Thus, the thicker the seagrass macrodetritus piles, the thicker the habitats of the earthworms ($Y = 11 + 0.89X$, $X$: the thickness of seagrass macrodetritus pile in cm; $Y$ = vertical range of *P. litoralis* in cm; figure 2). Under the northeast monsoon in winter, when seagrass macrodetritus accumulates on the north beach, its thickness can reach 30–50 cm. Most seagrass macrodetritus piles disappeared when the wind direction changed in summer; only thin layers of scattered seagrass macrodetritus were left on the beach. The earthworms could still be found either under these piles or in the sand in the moist part of the substrate. The earthworms originally scattered in seagrass piles were either carried along with seagrass to the sea or moved vertically downward to the sediment when the seagrass cover became increasingly thinner because of the wind.

In comparing the positions of the topmost and bottommost earthworms among 84 piles of seagrass macrodetritus of differing thickness, no difference was found between the day and night observations [topmost: $Y$ (day time) $= -7$ cm $+ 0.89 \times X$, $r = 0.90$, $p = 0.00$; $Y$ (night time) $= -7$ cm $+ 0.78 \times X$, $r = 0.85$, $p = 0.00$; bottommost: $Y$ (day time) $= -18$ cm $- 0.04 \times X$, $r = -0.08$, $p = 0.62$; $Y$ (night time) $= -20$ cm $+ 0.06 \times X$, $r = 0.15$, $p = 0.33$; $X$: thickness of seagrass piles; $Y$: vertical position of the earthworms; positive values indicate positions above the sand surface (figure 2). No earthworm was found more than 25 cm below the sand surface at Dongsha.

On the west coast of Taiwan, the earthworms were found in damp sand where only dry filamentous leaves of swamp oaks covered the surface. In addition, they were found near the very surface but remained covered by compact mangrove macrodetritus, where underlying layers were hypoxic. In these cases, the vertical ranges of *P. litoralis* were often less than 1 cm.

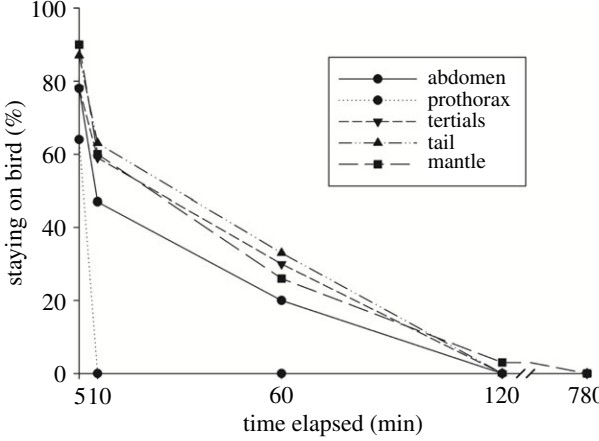

**Figure 3.** Percentage of *P. litoralis* earthworms remaining on different body parts of pigeons after placement.

**Table 3.** Test of independence between clinging capability of *P. litoralis* earthworms and the substrate type.

|  | seagrass | wood | $p\,(\chi^2)$ |
|---|---|---|---|
| clinging | 0 | 27 | <0.001 |
| non-clinging | 30 | 3 |  |

For earthworm cultures kept in the laboratory, light needed to be maintained or some earthworms would climb up the vertical walls of the culturing tanks. This behaviour suggests that the earthworms may surface in the evenings on the beaches, especially when the substrate is wet.

## 3.3. Clinging experiment (departure stage)

All 30 of the earthworms in the seagrass group dropped to the bottom of the beakers, while the seagrass remained suspended near the surface. The earthworms clearly could not cling to the seagrass leaves in seawater. In the wood-block group, 90% ($N = 30$) of the earthworms clung to the wood blocks; only three dropped to the bottom of the beakers. Their clinging capability was dependent on the substrate provided, that is, seagrass blades versus woodblocks ($p < 0.001$, $X^2$ test, table 3).

## 3.4. Birds as the vector (transfer stage)

Earthworms dropped from different body parts of the pigeons either by themselves or because of the cleaning behaviour of the birds soon after the beginning of the experiment (figure 3). For example, there was a higher than 40% chance of their staying on the pigeons for longer than 10 min, except when they were placed on the prothorax from which all dropped. Within 2 h, all except one earthworm, which was found dead on the mantle, had left the pigeons one way or another. In the light of these results, we did not test how the earthworms could board the pigeons at the various positions.

## 3.5. Seawater versus freshwater in the laboratory (transfer stage)

Small earthworms increased in size considerably faster than large earthworms, and a significant negative relationship was discovered in both treatments of different salinities; nevertheless, the linear relationships did not differ between freshwater and seawater (seawater: $Y\,(\%) = 98 - 295 \times X\,(g)$, $R^2 = 0.56$; freshwater: $Y\,(\%) = 130 - 379 \times X\,(g)$, $R^2 = 0.58$, analysis of covariance (ANCOVA), $p > 0.05$ in both comparisons of intercepts and slopes; $X$: initial weight of earthworms; $Y$: % weight increment; figure 4). Thus, no difference in weight increment between seawater and freshwater was found in the experiment lasting two months. Similar results were found one month after the treatments (K.S. 2013, unpublished data).

In this experiment, the survival of earthworms was discovered to be independent of the seawater and freshwater treatments at two-week, one-month and two-month intervals (table 4). By the end of the

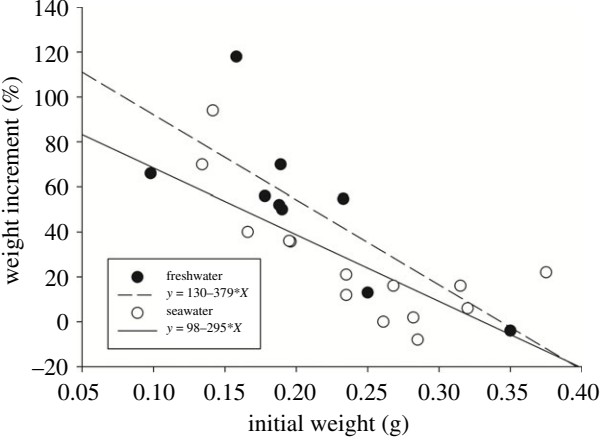

**Figure 4.** Comparison of the weight increment of the *P. litoralis* earthworm between seawater ($n = 14$) and freshwater ($n = 9$) after two months of treatment. The slopes of the two regression lines {seawater: $Y$ (%) $= [98 − 295 \times X$ (g)$] \times 100\%$, $R^2 = 0.56$, $p < 0.01$; freshwater: $Y$ (%) $= [130 − 379 \times X$ (g)$] \times 100\%$, $R^2 = 0.58$, $p < 0.01$} are not significantly different ($p = 0.31$, ANCOVA), and neither are the intercepts ($p = 0.40$, ANCOVA).

**Table 4.** Test of independence between salinity and mortality of *P. litoralis* earthworm.

| time elapsed | category | alive | dead | *p*-value ($\chi^2$) |
|---|---|---|---|---|
| two weeks | seawater | 24 | 6 | 0.58 |
| | freshwater | 13 | 2 | |
| one month | seawater | 15 | 15 | 0.29 |
| | freshwater | 10 | 5 | |
| two months | seawater | 14 | 16 | 0.40 |
| | freshwater | 9 | 6 | |

**Table 5.** Field test of independence between substrate type and mortality of *P. litoralis* earthworms ($p = 0.05$, Fisher's exact test between wood and seagrass; $p < 0.01$ in the $2 \times 3$ table).

| category/status | alive | dead | *p*-value |
|---|---|---|---|
| wood | 5 | 25 | — |
| seagrass | 0 | 30 | 0.05 |
| black cloth | 17 | 13 | <0.01 |

experiment, 47% ($N = 30$) and 60% ($N = 15$) of individuals had survived in the seawater and freshwater groups, respectively.

## 3.6. Survival in wood and seagrass (transfer stage)

In this field experiment lasting 53 days, 17% ($N = 30$) of the earthworms in the wood-block group survived, but none survived in the seagrass group ($N = 30$); thus, survival was dependent on the substrate ($p = 0.05$, Fisher's exact test). These results supported the hypothesis of floating wood being a potential vector in the sea. The surprising part of the result was that the highest survival rate of 57% was found in the black-cloth group (table 5).

## 4. Discussion

The abundance of *P. litoralis* in thick piles of seagrass macrodetritus on the north coast of Dongsha Island seemed to provide an opportunity for earthworm departure. This is because most macrodetritus that

accumulates along the coast is blown out to sea once the southwest monsoon prevailed in summer. However, this hypothesis of the seagrass vector for dispersal requires testing on several fronts.

One clear behaviour of this earthworm is light avoidance; therefore, they are unlikely to remain on the surface of water where seagrass leaves are. This was supported by both the vertical distribution of the earthworms in the field (figure 2) and laboratory observations. The distribution of the earthworms down to 20–25 cm deep in coral sand suggests that the sand layer may function as a sanctuary in the season when little seagrass macrodetritus exists on the surface of the beach. The earthworms could come to the surface at night, especially when it is wet, such as during rainy periods. It was unclear whether the earthworms were actually blown out to sea with seagrass macrodetritus.

The lack of ability of *P. litoralis* to cling to seagrass leaves certainly makes it unlikely for said leaves to serve as an effective vector in seawater. In fact, dead leaves were discovered to sink, presumably after the air in the lacuna had dissolved (K.S. 2011, personal observations). This certainly limits the range that seagrass leaves could travel in the sea because dead leaves on the bottom are unlikely to travel far. The seagrass hypothesis fails in both the departure and transfer stages of dispersal.

The bird hypothesis is undermined in the departure stage of dispersal because the earthworms are at least 3 cm deep in seagrass piles or sand. The earthworms have a greater chance of accessing birds if these birds nest on seagrass macrodebris. This behaviour was not observed on Dongsha Island but remains possible on uninhabited islands. When considering the transfer stage of dispersal, the bird hypothesis has an advantage because flying substantially shortens the duration required between two suitable but unconnected habitats.

The stowaway experiments on birds were performed under the assumption that the worms could find a way to enter the plumage of vector birds, such as when they nested on seagrass macrodetritus. The experiments, however, suggested that this potential was limited because most of the experimental worms fell off within a short time. However, egg capsules could still have the chance to attach to birds, but unfortunately, we did not observe any earthworm egg capsules in our investigations.

Ballast sand as a vector was first raised by Blakemore [4], and this hypothesis is attractive because of the ability of ballast to cross oceans easily. Moreover, the earthworms would stay in a habitat highly similar to their original ones when transferring. The departure stage of dispersal is easy to imagine when sand is collected, and the stowaway earthworms therein do not require any means of clinging to anything. The settlement stage, which begins upon reaching a new suitable habitat, is difficult to assess because modern ships use ballast water rather than sand [4]. Ballast sand may be depleted of food if on ships for too long. The exact site for releasing ballast sand is critical to the survival of stowaway earthworms because they have no protection in the water column. We see no reason for sailors to expend effort to return the sand to high water marks where suitable habitats of *P. litoralis* are usually found. Thus, the settlement stage may be the weak point of this hypothesis.

The current hypothesis is flexible in that no particular vector is specified. Regarding the departure stage of dispersal, our investigation did not support seagrass as a likely candidate. On the other hand, wood was supported by our clinging experiment. Because of both the mass of floating trunks and the crevices available, wood could be a long-term vessel because it provides both food and shelter for stowaway earthworms.

Our field experiment revealed that *P. litoralis* not only survived but also grew during their two-month immersion in seawater (figure 4). This euryhaline capability is certainly required during the transfer stage according to the current hypothesis, regardless of whether the real vector is driftwood or wood boats.

The euryhaline characteristic did not necessarily evolve to enable dispersal because it is also required for living in the intertidal zone. Long-distance travel may be a poor trait under individual selection because it is not adaptive to local environments; however, the situation is different at the species level. Once widespread in the oceans, the species is less likely to go extinct.

Cocoons are a stage at which dispersal could occur [43–45]. Because of their narrow width, *P. litoralis* could produce only small cocoons [4,46]. Small cocoons could be one of the reasons we were not successful in finding them in this study. Blakemore [4] suggested rafting of cocoons as a secondary dispersal mechanism of *P. litoralis* (with ballast sand being the primary) [4]. The ability of adults to survive long journeys at sea seems to make the role of cocoons in dispersal unnecessary. Yet, cocoons may, in fact, have more opportunities to depart natal habitats when attached to small floating objects that could easily move out to sea. Future research could test whether the transfer stage occurs mostly in the form of dormant cocoons or hatched individuals when cocoons are readily available.

More difficult to assess is whether driftwood or man-made wood boats have dispersed *P. litoralis* worldwide. Driftwood has added potential because it originates from land and may have to pass and remain in many freshwater habitats, where *P. litoralis* also occurs, before reaching the sea [9,38,42,47–49].

**Table 6.** Comparison of two potential vectors for *P. litoralis* involving wood at sea.

| trait | potential vector | |
|---|---|---|
| | wood boats | driftwood |
| duration of existence | Short: they appeared only when civilizations build wood boats/ships. | Long. Driftwood is a product of nature. |
| suitable habitats | Cracks on wood boats, near and above water lines. Cracks tend to be fixed during maintenance for seaworthy vessels. | Cracks and crevices occur naturally in driftwood. |
| boarding the vector (departure stage) | On beaches where boats land. Frequent departure and settlement. On wooden wharfs where boats dock. | On riverbanks where driftwood lands temporarily. Departure of driftwood may be much less frequent than that of wood boats. |
| potential distance of travel (transfer stage) | Humans decide. Small boats do not travel far, whereas large boats are less likely to be beached. | Current and wind decide where to land and thus the distance and direction of dispersal. |
| arriving at new habitat (settlement stage) | On beaches where boats land. | On beaches where driftwood lands. |
| uninhabited versus inhabited islands | Uninhabited islands are less likely to have boats. | Presence or not is unrelated to humans. |

This inland stage may well be the time at which earthworms attach to driftwood. Moreover, beached wood may drift again (departure) if it is not pushed high enough in the intertidal zone.

Wooden vessels have been proposed as a possible vector for this earthworm. They differ from driftwood in terms of 'docking' sites and frequencies. Boats, especially large ones, are more likely to dock in harbours rather than landing on beaches. This may explain why docks are known to have this earthworm [5]. Boats are also much faster and more directional than driftwood. Large vessels are known to have travelled around the world during the Age of Exploration. They had the potential to have served as a super vector for *P. litoralis*. On the other hand, protective coatings are often applied to wood boats, and thus, only cracks—and presumably deep ones—are suitable for light-avoiding earthworms.

Comparing the two potential vectors of the earthworm, the probability of departure on a wood boat was much higher than that on a piece of driftwood, but it is unclear which type prevailed in terms of number (table 6). The transfer stage is much more efficient by wood boat than by driftwood because boats have a destination, which is likely to be close to a suitable habitat for the earthworms. The settlement stage is similar for the two vectors except wood boats could also use wooden piers, which is unlikely for driftwood. Thus, wood boats may have advantages in all three stages of dispersal.

Wooden vessels have existed for only a short time on an evolutionary time scale. However, this may well have been long enough to disperse *P. litoralis* around the world as well as maintain the genetic connectivity of the populations. However, few wooden vessels are capable of long-distance journeys today; therefore, the effective dispersal vectors may be virtually 'extinct'. The extent of gene flow among these earthworm populations may thus be substantially lower relative to the past few centuries under this scenario, and divergence among populations of *P. litoralis* worldwide is expected.

Ethics. The use of the pigeons was approved by the IACUC of National Sun Yat-sen University.

Data accessibility. The data are provided in electronic supplementary material [50].

Authors' contributions. Conceptualization and methodology: S.-Y.C., K.S. Data curation and Investigation: S.-Y.C. Formal analysis: S.-Y.C., C.-H.H. Funding acquisition, project administration, resources and supervision: K.S. Validation, writing original draft, writing review and editing: C.-H.H., K.S. Visualization: S.-Y.C., C.-H.H.

Competing interests. We declare we have no competing interests.

Funding. We received no funding for this study.

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
