## [Peer Review File · Royal Society Open Science]

Review History

RSOS-202297.R0 (Original submission)

Review form: Reviewer 1

Is the manuscript scientifically sound in its present form?

No

Are the interpretations and conclusions justified by the results?

Yes

Is the language acceptable?

Yes

Do you have any ethical concerns with this paper?

No

Have you any concerns about statistical analyses in this paper?

No

Recommendation?

Accept with minor revision (please list in comments)

Comments to the Author(s)

One very important consideration in the dispersal of this species is to ask how many species are we talking about. There is some clear genetic differentiation within nominal *P. litoralis*, so the idea that it has a global distribution is premature. There is some good work done in Thailand on this question. Seesamut, T., Jirapatrasilp, P., Chanabun, R., Oba, Y. & Panha, S. Size variation and geographical distribution of the luminous earthworm *Pontodrilus litoralis* (Grube, 1855) (Clitellata, Megascolecidae) in Southeast Asia and Japan. *Zookeys*. 862, 23–43. <https://doi.org/10.3897/zookeys.862.35727> (2019).

Seesamut, T., Sutcharit, C., Jirapatrasilp, P., Chanabun, R. & Panha, S. Morphological and molecular evidence reveal a new species of the earthworm genus *Pontodrilus* Perrier, 1874 (Clitellata, Megascolecidae) from Thailand and Peninsular Malaysia. *Zootaxa* 4496, 218–237. <https://doi.org/10.11646/zootaxa.4496.1.18> (2018).

I suggest that the authors review this evidence, and DNA barcode data from the genus in global databases. They still have a good set of results regarding potential dispersal mechanisms but it is not necessarily relevant to a large-scale distribution. That depends of it actually being really large scale. With all due respect to Blakemore, he is quick to say there is no difference when in fact there are differences. He has been wrong about some other species in his lumping things together as synonyms, even overlooking very clear morphological variation. We also know from many terrestrial earthworms that there is cryptic variation: species that are in fact composed of genetically separate lineages while remaining morphologically very similar or identical. Semi-aquatic to aquatic earthworms typically have broader distributions and apparently have better dispersal ability, so I remain open to the possibility that *P. litoralis* (whatever that really is) does have a large distribution. But it also has local genetic variation. I have even found them in forest soils well above sea level in the Virgin Islands, suggesting that the species has ecological variants or a wide niche range.

I suggest that some of the conclusions could be modified in light of this information about the genetic and other diversity of *Pontodrilus*.

The type locality of the species is in France, on the Mediterranean shore. This is the first discovery location as cited in Grube 1855 (ref 11). So far nobody has been able to collect new specimens from a point near the type locality.

Decision letter (RSOS-202297.R0)

Dear Dr Soong

On behalf of the Editors, we are pleased to inform you that your Manuscript RSOS-202297 "How to cross the sea: Testing the dispersal mechanisms of the cosmopolitan earthworm *Pontodrilus litoralis*" has been accepted for publication in Royal Society Open Science subject to minor revision in accordance with the referees' reports. Please find the referees' comments along with any feedback from the Editors below my signature.

Please submit your revised manuscript and required files (see below) no later than 7 days from today's (ie 14-Jul-2021) date. Note: the ScholarOne system will 'lock' if submission of the revision is attempted 7 or more days after the deadline. If you do not think you will be able to meet this deadline please contact the editorial office immediately.

on behalf of Prof Pete Smith (Subject Editor)
openscience@royalsociety.org

Reviewer comments to Author:
Reviewer: 1

Comments to the Author(s)

One very important consideration in the dispersal of this species is to ask how many species are we talking about. There is some clear genetic differentiation within nominal *P. litoralis*, so the idea that it has a global distribution is premature. There is some good work done in Thailand on this question. Seesamut, T., Jirapatrasilp, P., Chanabun, R., Oba, Y. & Panha, S. Size variation and geographical distribution of the luminous earthworm *Pontodrilus litoralis* (Grube, 1855) (Clitellata, Megascolecidae) in Southeast Asia and Japan. *Zookeys*. 862, 23–43. <https://doi.org/10.3897/zookeys.862.35727> (2019).

Seesamut, T., Sutcharit, C., Jirapatrasilp, P., Chanabun, R. & Panha, S. Morphological and molecular evidence reveal a new species of the earthworm genus *Pontodrilus* Perrier, 1874 (Clitellata, Megascolecidae) from Thailand and Peninsular Malaysia. *Zootaxa* 4496, 218–237. <https://doi.org/10.11646/zootaxa.4496.1.18> (2018).

I suggest that the authors review this evidence, and DNA barcode data from the genus in global databases. They still have a good set of results regarding potential dispersal mechanisms but it is not necessarily relevant to a large-scale distribution. That depends of it actually being really large scale. With all due respect to Blakemore, he is quick to say there is no difference when in fact there are differences. He has been wrong about some other species in his lumping things

together as synonyms, even overlooking very clear morphological variation. We also know from many terrestrial earthworms that there is cryptic variation: species that are in fact composed of genetically separate lineages while remaining morphologically very similar or identical. Semi-aquatic to aquatic earthworms typically have broader distributions and apparently have better dispersal ability, so I remain open to the possibility that *P. litoralis* (whatever that really is) does have a large distribution. But it also has local genetic variation. I have even found them in forest soils well above sea level in the Virgin Islands, suggesting that the species has ecological variants or a wide niche range.

I suggest that some of the conclusions could be modified in light of this information about the genetic and other diversity of *Pontodrillus*.

The type locality of the species is in France, on the Mediterranean shore. This is the first discovery location as cited in Grube 1855 (ref 11). So far nobody has been able to collect new specimens from a point near the type locality.

===PREPARING YOUR MANUSCRIPT===

===PREPARING YOUR REVISION IN SCHOLARONE===

Author's Response to Decision Letter for (RSOS-202297.R0)

See Appendix A.

Decision letter (RSOS-202297.R1)

Dear Dr Soong,

I am pleased to inform you that your manuscript entitled "How to cross the sea: Testing the dispersal mechanisms of the cosmopolitan earthworm *Pontodrilus littoralis*" is now accepted for publication in Royal Society Open Science.

on behalf of Pete Smith (Subject Editor)

Appendix A

Dear Editor and Reviewer,

We already followed your comments to revise our article and make it better. Please have a check. We are looking forward for your good news.

Best regards,

Keryea Soong, Ph.D.
Department of Oceanography, National Sun Yat-sen University
PI, Dongsha Atoll Research Center
07.17.2021